# Facile Charge Transfer between Barbituric Acid and Chloranilic Acid over g-C$_3$N$_4$: Synthesis, Characterization and DFT Study

**Gaber A. M. Mersal** [1,*]**, Mohamed M. Ibrahim** [1]**, Mohammed A. Amin** [1]**, Amine Mezni** [1]**, Nasser Y. Mostafa** [2]**, Sarah Alharthi** [1]**, Rabah Boukherroub** [3] **and Hamdy S. El-Sheshtawy** [4,*]

[1] Department of Chemistry, Faculty of Science, Taif University, Taif 888, Saudi Arabia; ibrahim@tu.edu.sa (M.M.I.); mohamed@tu.edu.sa (M.A.A.); aminemrezni@yahoo.fr (A.M.); world-sososo-2007@hotmail.com (S.A.)
[2] Chemistry Department, Faculty of Science, Suez Canal, Ismailia 41522, Egypt; nmostafa@yahoo.com
[3] Univ. Lille, CNRS, Centrale Lille, ISEN, Univ. Valenciennes, UMR 8520—IEMN, 59000 Lille, France; rabahboukherroub@univ-lille1.fr
[4] Chemistry Department, Faculty of Science, Kafrelsheikh University, Kafrelsheikh 33516, Egypt
* Correspondence: gamersal@tu.edu.sa (G.A.M.M.); hamdyalfy@gmail.com (H.S.E.-S.)

**Abstract:** The molecular complexes between barbituric acid (BU) and chloranilic acid (ChA) over graphitic nitride (g-C$_3$N$_4$) are investigated. The molecular complexes and the nanocomposite were investigated both in solid state and in methanol. The solid complexes and the corresponding nanocomposite were investigated using FTIR, TGA, and UV-Vis spectroscopy. The structures were explored using DFT calculations using wB97XD/ and def2-TZVP basis set. The DFT calculations revealed the formation of hydrogen-bonded complexes, which initiate the proton transfer from ChA to BU. Immobilization of the BUChA complex over the g-C$_3$N$_4$ sheet was stabilized by weak non-covalent interactions, such as π–π interactions. g-C$_3$N$_4$ facilitated the charge transfer process, which is beneficial for different applications.

**Keywords:** barbituric acid; chloranilic acid; charge transfer; g-C$_3$N$_4$; DFT calculation

## 1. Introduction

The charge transfer process (CT) has been extensively investigated in material science [1–4], biological systems [5,6], and crystal engineering [7,8]. The CT process is initiated between high electron donor molecules (Donors) and electron-deficient molecules (Acceptors) to form stable CT adducts [9,10]. The CT adducts are stabilized by various non-covalent forces, such as hydrogen bonding [11–14], halogen bonding [15–17], and π–π interactions [18,19]. The CT process is initiated by HB formation and proceeds through the proton transfer process (PT) [20–22]. Different organic acceptors efficiently act as proton donors, such as chloranilic acid (ChA) [13,20,21], 7,7,8,8-tetracyanoquinodimethane (TCNQ) [1], and 2,3-dichloro-5,6-dicyano-1,4-benzoquinone (DDQ) [21]. The kinetics of the PT process play a key role in different biological systems [23–25] and electronic devices [26,27].

Metal-free graphitic nitride (g-C$_3$N$_4$) has been widely used for photocatalytic applications [28], such as hydrogen production [29,30], pollutants removal [31], and li-ion batteries [32–34]. The superior use of g-C$_3$N$_4$ is attributed to the substrate's low-cost, easy synthesis, excellent visible light response, and high surface to volume ratio. g-C$_3$N$_4$ is used for constructing Z-scheme photocatalysis to mediate the interfacial charge transfer process [35,36]. The small band gap of g-C$_3$N$_4$ (~2.7 eV) leads to visible light absorption; however, the fast electron/hole recombination process due to the low charge mobility limits the photocatalytic activity. Immobilization of organic molecules over g-C$_3$N$_4$ surface facilitates the charge transfer process and inhibits the recombination process [37]. For example, introducing 11,11,12,12-tetracyanonaphtho-1,4-quinodimethane (TCNAQ) over g-C$_3$N$_4$

surface enhanced the charge separation and induced charge carriers migration [38]. This process inhibits the charge recombination and enhances photocatalytic activity. Constructing g-C$_3$N$_4$/g-C$_3$N$_4$ heterojunction facilitates charge separation and prolongs the charge carrier's lifetime, which enhances photocatalytic activity [39]. Prasaanth et al. prepared an efficient photocatalyst from cellulose nanofiber foam embedded g-C$_3$N$_4$ to enhance the charge carrier's separation [40]. The interfacial charge transfer process was achieved by the π–π stacking of perylene diimide/g-C$_3$N$_4$ heterojunction [41]. Barbituric acid was used for non-metal doping of g-C$_3$N$_4$ due to the similar structure of melamine to form the C–C bond during thermal polymerization [42–44]. Although there is considerable work on non-metal doping of g-C$_3$N$_4$ with organic ligands, the donor–acceptor pair/g-C$_3$N$_4$ heterojunction has been scarce in literature [37].

In this work, we investigated the molecular complexes between BU (donor) and ChA (acceptor) both in the absence and in the presence of g-C$_3$N$_4$ in methanol and in a solid state. The composite was investigated by different methods, such as FTIR, TGA, and UV-Vis spectroscopy. In addition, DFT calculations using the wB97XD method, which accounts for the weak dispersion forces coupled with the def2-TZVP basis set, were used to explore the structure and the binding forces. The interfacial charge transfer process in BUChA and BUChA/g-C$_3$N$_4$ composites was investigated by UV-Vis spectroscopy and confirmed by DFT calculations. These results are beneficial for constructing the heterojunction organic/g-C$_3$N$_4$ systems for photocatalytic applications.

## 2. Materials and Methods

### 2.1. Materials and Characterization

Both BU and ChA were purchased from Sigma-Aldrich, Germany, while the solvents were analytical grades and used without further purifications. Fresh solutions (1 mM) were prepared from both the donor (BU) and acceptor (ChA) by dissolving the appropriate amount of the solute into the solvent volume. In order to determine the formation constant ($K_{CT}$) and molar absorptivity ($\varepsilon$), successive concentrations (0–1.2 mM) of the acceptor (ChA) were added to 2.8 mLof BU (0.05 mM) methanol solution at 25 °C. The UV-Vis electronic absorption of the complexes was recorded with an Agilent UV-NIR spectrophotometer in the range from 200 to 800 nm in a 1 cm quartz cuvette. FTIR measurements were recorded with JASCO spectrometer (FTIR-4100, JASCO, Tokyo, Japan). The TGA thermal measurements were recorded using Shimadzu (DTG 60H) under nitrogen in the range from 10 to 1000 °C. g-C$_3$N$_4$ morphology was measured using SEM (VP-SEM 20KV, Hitachi, Japan).

### 2.2. Preparation of Solid Complexes and Nanocomposite

g-C$_3$N$_4$ was prepared by heating 5 g urea in a covered crucible at 550 °C for 3 h in air with a ramping rate of 10 °C/min [45]. BUChA CT solid complexes were prepared by mixing BU (0.55 mmol in 25 mLof MeOH) and ChA (0.55 mmol in 25 mLof MeOH) under continuous stirring (1 h). A deep-purple color of the BUChA complex was observed with time. The solutions were kept at room temperature overnight for slow evaporation. Immobilization of the BUChA complex over the g-C$_3$N$_4$ was prepared by mixing the BUChA solution (100 mg in 25 mLof MeOH) and x-g-C$_3$N$_4$ (x = 5 mg and 10 mg in 25 mL of MeOH for 5% and 10% wt/wt, respectively). The solution was stirred for 3 h at room temperature and left overnight for solvent evaporation. The obtained solid product (x-g-C$_3$N$_4$/BUChA) was washed with ethanol three times to remove the non-bonded BUChA complex. The purified solid was further used for different characterizations.

### 2.3. Computational Details

Density functional theory (DFT) through wB97XD/def2-TZVP method was used to study the molecular forces for the interaction between BU and ChA. The structures of the individual molecules and the complexes were optimized by the Gaussian 09 program [46]. Frequency calculations were performed on the optimized structures to ensure that the

structures had minimum energies on the potential energy surface, which was confirmed by the positive vibration modes. The solvent effect on the CT complexes was performed using the self-consistent reaction field (SCRF) method and polarized continuum model (PCM). Charge population and charge transfer analysis were calculated using the Multiwfn program [47]. The g-$C_3N_4$ nanosheet was constructed based on the (011) lattice plane supercell ($2 \times 2 \times 1$), according to the literature [48]. The terminal N atoms were truncated with H atoms for layer stability [49].

## 3. Results and Discussion

### 3.1. Spectroscopic Properties and Structure Characterization of BUChA

The absorption spectra of BU showed a weak peak at 320 nm, whereas ChA showed a characteristic peak at 460 nm (Figure 1a). Upon mixing BU and ChA, the absorption peak of the complex (BUChA) was observed at 570 nm (Figure 1a). Figure 1b shows the electronic absorption spectra of the successive addition of ChA to the BU solution at 25 °C. By adding the methanolic solution of ChA to the BU solution, a new absorption peak at 570 nm was observed in addition to the ChA peak at 370 nm. The color of the solution changed to purple with the ChA concentration increase. The appearance of the new broad peak at 570 nm at which neither BU nor ChA absorb was attributed to the stable charge-transfer complexes' formation through radical anion formation [50,51]. The stoichiometric of the interaction between BU and ChA was determined by the Jobs method through monitoring the absorption spectra at the new 570 nm peak [52]. The deflection point at the 0.5 molar ratio indicated the 1:1 complex formation of BU and ChA in methanol (data not shown).

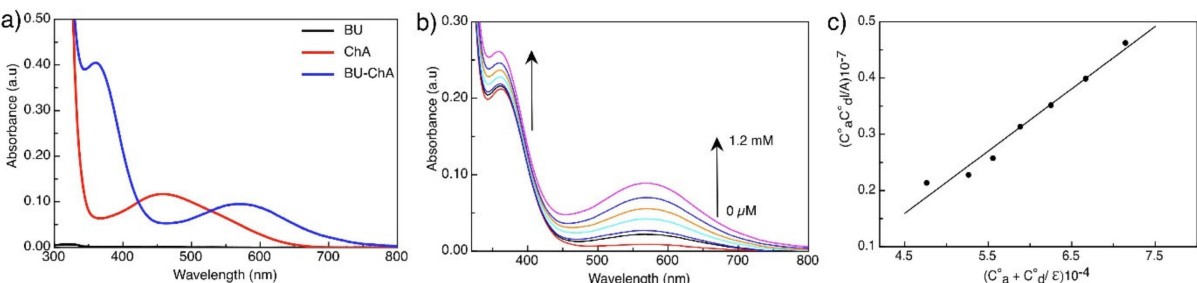

**Figure 1.** (**a**) absorption spectra of BU (0.05 mM), ChA (0.05 mM), and BUChA complex, (**b**) titration of BU (0.05 mM) and different concentration of ChA in methanol, (**c**) modified Benesi–Hildebrand plot for the BU and ChA titration in methanol.

The characteristic parameters for the BUChA complexes, such as formation constant ($K_{CT}$), and molar absorptivity ($\varepsilon_{CT}$) in methanol, were determined using Equation (1) [53,54]:

$$\frac{C_a^0 C_d^0 l}{A} = \frac{1}{K\varepsilon} + \frac{C_a^0 + C_d^0}{\varepsilon} \tag{1}$$

$C_a$ and $C_d$ were the initial concentration of both acceptor and donor, respectively. Figure 1b shows the straight-line plot, which confirms the 1:1 complex formation in methanol. The calculated $K_{CT}$ and $\varepsilon_{CT}$ were $2.8 \times 10^3$ and $2.4 \times 10^3$, respectively (Table 1), which indicate the moderate strength of the complex.

**Table 1.** Measured formation constant ($K_{CT}$, M$^{-1}$), extinction coefficient ($\varepsilon_{CT}$, M$^{-1}$), calculated HOMO energies, LUMO energies, energy gap, and Hirshfeld atomic charges for the BUChA adduct.

| System | $K_{CT}$ | $\varepsilon_{CT}$ | $E_{HOMO}$ | $E_{LUMO}$ | $E_{gap}$ | C/e | |
|---|---|---|---|---|---|---|---|
| | | | | | | **BUChA-I** | **BUChA-II** |
| BU | – | – | −4.04 | −1.05 | 2.99 | 0.0 | 0.0 |
| BUChA | $4.2 \times 10^3$ | $3.5 \times 10^5$ | −4.10 | −1.96 | 2.36 | +0.21 | +0.13 |
| BUChA-g-$C_3N_4$ | – | – | −5.13 | −2.9 | 2.23 | +0.18 | - |

The solid complex BUChA was investigated by thermal analysis and FTIR spectroscopy. TGA-DSC of BU showed three weight loss steps. The total weight loss of 94% at 900 °C was observed, indicating that the final residue was not carbonaceous (Figure 2a). BU realized a small weight loss of 3.42%, probably due to the evaporation of bound surface adsorbed moisture. The second step showed a significant weight loss with an exothermic peak at 208 °C in the range of 150–250 °C, due to the loss of 1.5 molecules of CO (found: 31.07; calc.: 32.81%) [55]. The second step showed a weight loss of 18.17 (calc. 18.75 %) in the range of 240 to 350 °C, due to the elimination of 1.5 molecules of $CH_4$. The third step started below 500 °C, assigned to the loss of two $N_2$ molecules (found: 43.12; calc.: 43.57%). The TGA-DSC thermogram of the CT complex BUChA showed the decomposition interaction started with an exothermic peak at 221.50 °C, indicating its higher stability than that of the pristine BU (Figure 2b). This decomposition step was accompanied by a weight loss of 36.47% (calc. 37.38%), which may be attributed to the elimination of $4H_2O + CO + CO_2$. The second endothermic peak occurred at 243.9 °C and was accompanied by a weight loss of 16.52% (17.80%), which may be due to the loss of $CO_2 + CH_4$. The final step started below 400 °C, was assigned to the removal of the remaining nitrogenous residues.

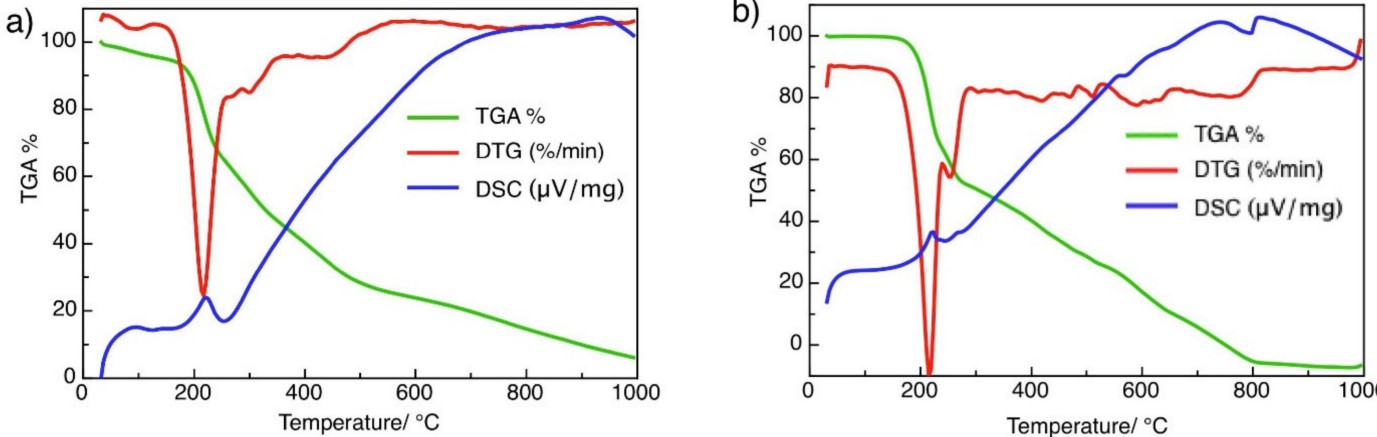

**Figure 2.** TGA plot of (**a**) BU and (**b**) BUChA adduct.

The interaction between BU and ChA was confirmed by the solid FTIR measurements. Figure 3a shows the individual FTIR spectra of BU, ChA, and the BUChA complex. The BU structure showed the presence of the $\nu$(OH) at 3476 $cm^{-1}$, $\nu$(N–H) 3214 $cm^{-1}$, and $\nu$(C=O) at 1750 $cm^{-1}$ [37]. The BUChA complex showed the combination of both BU and ChA with the decrease in the peaks' intensities and slight changes in peak position. For example, the intensity of the peaks $\nu$(OH) at 3476 $cm^{-1}$ and $\nu$(N–H) at 3214 $cm^{-1}$ was broaden and slightly declined to 3450 $cm^{-1}$ and $\nu$(N–H) at 3190, respectively, which was attributed to the engagement of the two functional groups in hydrogen bonding [50,51]. In addition, the $\nu$(C–Cl) at 845 $cm^{-1}$ in BUChA confirmed the C–Cl···H HB formation (Figure S1).

In order to investigate the molecular forces for interactions between BU and ChA, DFT calculations using wB97XD/def2-TZVP level of theory were used. Figure 4 (Tables S1–S4) shows the optimized structures of the most stable BU structure [56] and ChA. The calculated molecular electrostatic surface potential (MESP) of BU showed the presence of negative electrostatic potential ($V = -31.4$ and $-25.1$ kcal/mol) over the carbonyl (C=O) groups, while positive electrostatic potential ($V = +40.1$ and $+31.4$ kcal/mol) on -OH and –NH groups, respectively [37]. On the other hand, the ChA acceptor shows less negative electrostatic potential on –C=O groups ($V = -22.6$ kcal/mol) and low positive potential on –Cl and –OH groups, which nominated ChA for accepting electrons. Different orientations of the BUChA (BUChA-I-IV) complex were explored to explore the possible binding between BU and ChA. Figure 5a (Figure S3 and Tables S5 and S6) shows the most stable optimized structure of the molecular complex BUChA-I where short contact hydrogen bonding (HB) between the two molecules stabilized the complex formation. The HB in

O7···H15–O14 showed a 1.80 Å bond distance and 148° bond angle, which was 33.8% less than the sum of vdW radius. The short bond distance initiated the proton transfer from ChA to BU to form ChA$^-$···BU$^+$ species. The structure was further stabilized by another weak HB (C2H12···O18, 2.40 Å and 131° bond angle). The binding process could occur through the isomer in Figure 5b (BUChA-II, (Tables S7 and S8), where short HB (bond distance 1.75 and bond angle 153°) in addition to H10···Cl27 HB (bond distance 2.49 Å and bond angle 148°). However, the BUChA-I structure in Figure 5a was more stable by 2.4 kcal/mol than the BUChA-II structure. The solvent effect-using methanol ($\varepsilon$ = 32.6) was performed using the SCRF and PCM on the optimized structures of BUChA-I. The results showed the stabilization of the complexes on the methanol (6.9 kcal/mol), attributed to higher charge separation in the excited state [57]. The electronic absorption spectrum of BUChA in methanol was predicted using DT-DFT calculations. Figure S4 shows the presence of the CT band at 570 nm, which was consistent with the experimental CT band, with oscillator strength (*f*) of 0.012, which was assigned to HOMO $\rightarrow$ LUMO transition (98%). The HOMO orbitals of the BUChA-I position on both BU and ChA, while only ChA contributed to the LUMO orbitals of the complex (Figure 5c,d), which indicated the $\pi \rightarrow \pi^*$ electronic transition from the HOMO of BU to the ChA LUMO.

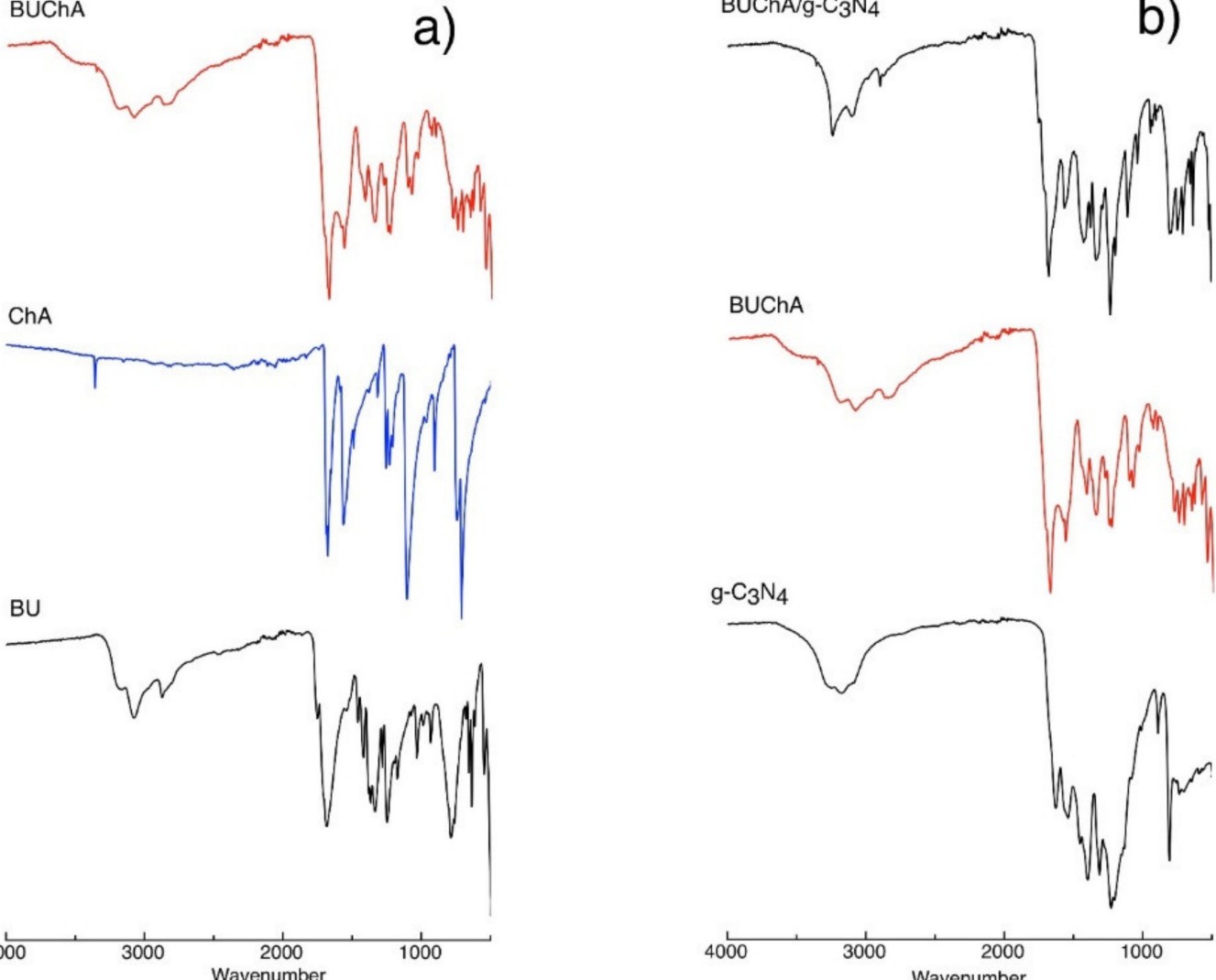

**Figure 3.** FTIR for the interaction between (**a**) BU and ChA and (**b**) immobilization of BUChA over g-C$_3$N$_4$ nanosheet.



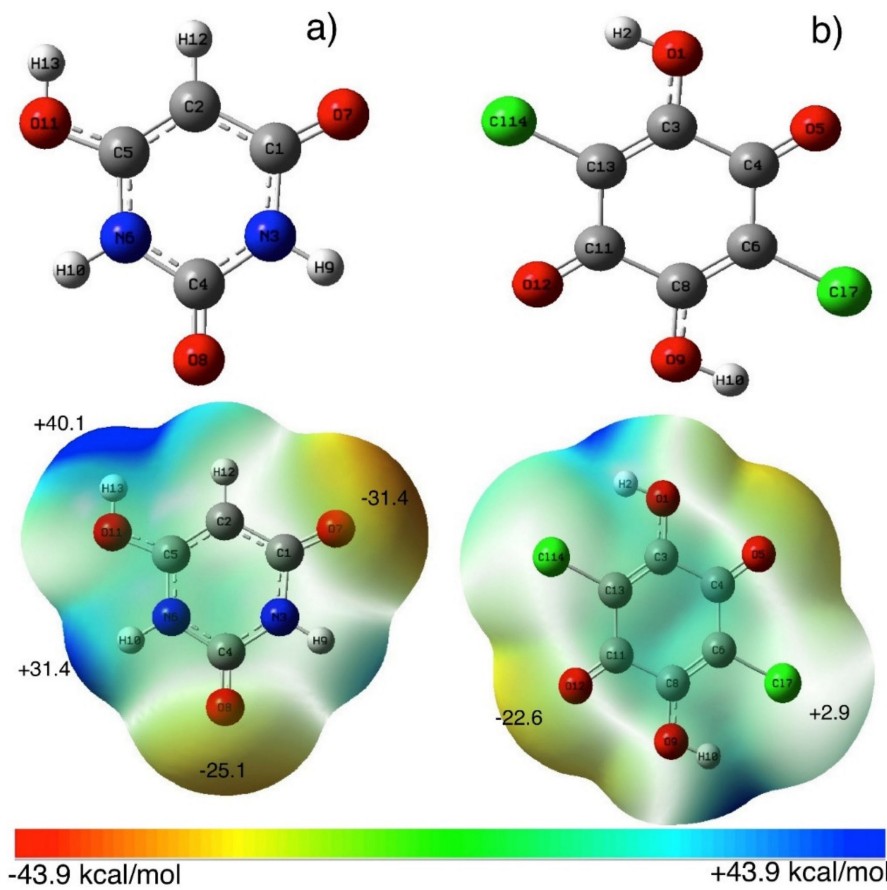

**Figure 4.** Optimized structure and MESP of (**a**) BU and (**b**) ChA in the gas phase calculated at wB97XD/def2-TZVP level of theory.

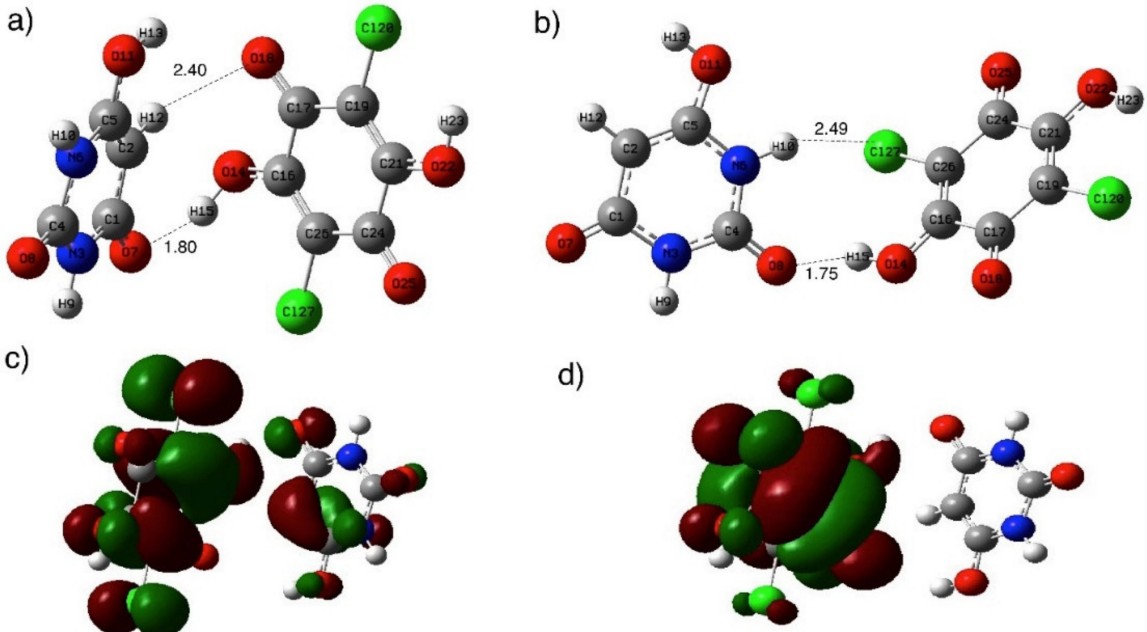

**Figure 5.** Optimized structure of (**a**) BUChA-I adducts and (**b**) BUChA-II, (**c**) HOMO orbitals and (**d**) LUMO orbitals of BUChA-I in the gas phase calculated at wB97XD/def2-TZVP level of theory.

### 3.2. Immobilization of BUChA over g-C$_3$N$_4$

Figure 6a shows the XRD pattern of g-C$_3$N$_4$ sheets. The g-C$_3$N$_4$ showed in-plane interlayer stacking and structural packing peaks at 26.5° and 12.7°, respectively [58]. Upon surface immobilization of BUChA on g-C$_3$N$_4$, the XRD pattern did not change; however, the interlayer stacking peak at 26.6° slightly broadened due to the strong interaction with the complex [44]. The SEM and TEM of the prepared g-C$_3$N$_4$ morphology indicated the formation of layered and sheet-like structures (Figure 6b,c) [37,44]. The FTIR spectra of g-C$_3$N$_4$ (Figure 3b) showed the characteristic s-triazine peak at 812 cm$^{-1}$, C–N vibration frequency at 1530 cm$^{-1}$, and the OH frequency at the 3000–3400 cm$^{-1}$ region [59]. The FTIR spectra of BUChA/g-C$_3$N$_4$ showed the combination of the frequency vibrations of BUChA and g-C$_3$N$_4$. However, the immobilization of BUChA over g-C$_3$N$_4$ stabilized the HB formation, which was shown by the intense peak of $\nu$(N–H) at 3214 cm$^{-1}$ (Figure 3b and Figure S2). The DFT optimized structure of BUChA/g-C$_3$N$_4$ nanosheets showed a stable structure by non-covalent interactions. Upon immobilization of the BUChA over the g-C$_3$N$_4$ nanosheets, the nanosheet facilitated the charge transfer process by decreasing the bond distance of the O7···H15–O14 HB from 1.80 to 1.72 Å (Figure 7 and Table S9) [60,61]. An extra stabilization of the BUChA complex occurred through $\pi$–$\pi$ interactions, such as O22···N39, C19···N97, and C2···N8, with bond lengths 2.84, 2.98, and 3.2 Å, respectively (Figure 7b). In addition, the calculated energy gap of BUChA/g-C$_3$N$_4$ (Table 1) was lower than BUChA, which attributed to the interfacial charge transfer process.

UV-Vis absorption spectroscopy was used to explore the immobilization of BUChA over g-C$_3$N$_4$ sheets. As discussed, ChA complex with BU showed a maximum absorption peak at 560 nm, which was ascribed to the radical anion (ChA$^-$•) formation upon interaction with the electron donor (BU). The intensity of the radical anion peak increased with time and reached the maximum intensity after 55 min (Figure 6d). However, the presence of g-C$_3$N$_4$ destabilized the radical anion formation and increased the interaction rate on the surface (Figure 6d–f) [37]. The radical destabilization on g-C$_3$N$_4$ could be attributed to the fast charge transfer formation over the nanosheets. This result revealed the easy charge transfer process over g-C$_3$N$_4$, which is favorable in photocatalytic applications [38].

Charge transfer analysis using Hirshfeld atomic population charges models was used. Table 1 shows the calculated Hirshfeld atomic charges on BU, ChA, BUChA, and BUChA/g-C$_3$N$_4$ composite. The individual donor (BU) and acceptor (ChA) were neutral molecules. Upon the formation of the BUChA adduct, the BU carried a positive charge (BUChA-I = +0.21 and BUChA-II = +0.13) and negative charges accumulated on ChA surface (−0.21), which indicated that the former molecule was the donor and the later was the electron acceptor. In addition, the higher charge transfer amount from BU to ChA in BUChA-I confirmed its stability against BUChA-II. In the BUChA-I/g-C$_3$N$_4$ composite, charge transfer (+0.18) was observed from the g-C$_3$N$_4$ to the BUChA adduct, which indicated that the former was the electron donor material [37]. The interfacial charge transfer process inhibited the charge carrier's (electron/hole) recombination on g-C$_3$N$_4$ surface, which is beneficial for photocatalytic applications [38,42–44].

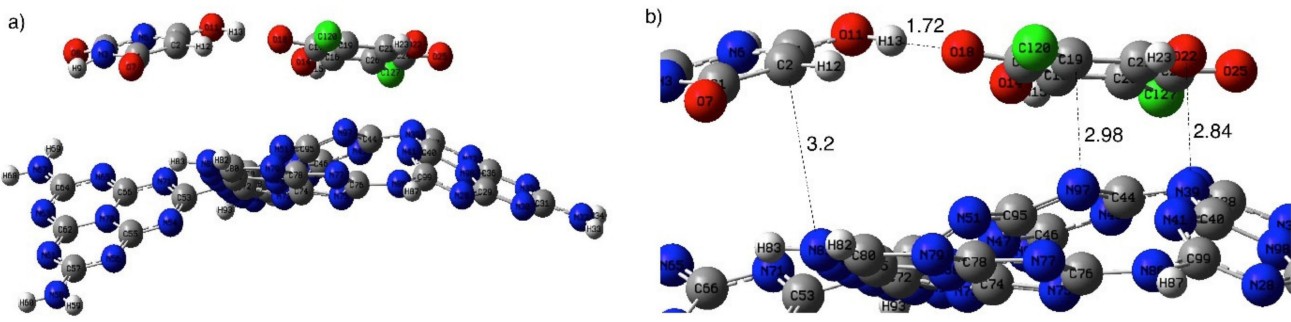

**Figure 6.** (**a**) XRD pattern, (**b**) SEM of g-$C_3N_4$, (**c**) TEM of the g-$C_3N_4$ nanosheet, (**d**) effect of time on the absorption spectra of BUChA, (**e**) effect of time on BUChA/5% g-$C_3N_4$, and (**f**) Time-absorbance relation in methanol.

**Figure 7.** Optimized structure of (**a**) BUChA/g-$C_3N_4$ and (**b**) enlarged BUChA/g-$C_3N_4$ structure in the gas phase.

## 4. Conclusions

The molecular adducts between BU and ChA were explored by means of experimental and theoretical techniques. A stable adduct (BUChA) was formed and gave a radical cation peak at 560 nm in methanol. The formation constant for the BUChA adduct was determined ($4.2 \times 10^3$), which indicates the moderate strength of the complex stability. The stability forces for the BUChA were explored by DFT calculations (wB97XD/def2-TZVP), which showed that hydrogen bonding is the main stabilizing force. Immobilization of the BUChA adduct over $g$-$C_3N_4$ nanosheets stabilized the proton transfer process and facilitated the charge transfer process, which is confirmed by a Hirshfeld charge transfer analysis. The facile charge transfer process inhibited the electron/hole recombination on $g$-$C_3N_4$ surface, which is beneficial for photocatalytic applications.

**Supplementary Materials:** The following are available online at https://www.mdpi.com/article/10.3390/cryst11060636/s1, Figure S1: The selected area of FTIR of BU and ChA, Figure S2: FTIR of BUChA over $g$-$C_3N_4$ nanosheet, Figure S3: optimized structure in the gas phase calculated at wB97XD/def2tzvpp level of theory, Figure S4: Calculated electronic spectra of BU-ChA complex in methanol using TD-DFT method. Table S1: the optimized structure of BU, Table S2: Calculated Frequencies of BU, Table S3: the optimized structure of CHL, Table S4: Calculated Frequencies of ChA, Table S5: the optimized structure of BUChA-I, Table S6: Calculated Frequencies of BUChA-I; Table S7: the optimized structure of BUChA-II, Table S8: Calculated Frequencies of BUChA-II, Table S9: the optimized structure of BUChA-I/$g$-$C_3N_4$.

**Author Contributions:** G.A.M.M. and H.S.E.-S. conceived and designed the experiments; M.M.I., A.M. and M.A.A. performed the experiments and analyzed the date; N.Y.M. and S.A. wrote the paper, R.B. and H.S.E.-S. writing review and supervision. All authors have read and agreed to the published version of the manuscript.

**Funding:** This work was financially supported by the High Attitude Research Center, Taif University, KSA Project Number: 1/440/6172.

**Institutional Review Board Statement:** Not applicable.

**Informed Consent Statement:** Not applicable.

**Data Availability Statement:** Data are contained within the article or Supplementary Material.

**Conflicts of Interest:** The authors declare no conflict of interest.

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
