# Peer review of "Facile Charge Transfer between Barbituric Acid and Chloranilic Acid over g-C3N4: Synthesis, Characterization and DFT Study"

_crystals, doi:10.3390/cryst11060636_

Round 1

Reviewer 1 Report

Comments

In general paper contains information which could be published. In present form I have many questions as Authors didn’t include many crucial information. There is no any proof that adduct was immobilized and this a main aspect of the manuscript. Moreover presented data are sometimes in contrary to Authors claims and do not agree with presented spectra. Many aspects must be rethink by Authors and maybe some new calculations will be necessary. Thus in my opinion manuscript requires major revision before could be accepted. In present form I suggest to reject the manuscript. Authors should improve also the grammar and other language problems (for example $ 2.1, 3.1 and Fig. 1 captions starts with small letters).

Please give information which is missing:

  1. In what conditions C3N4 was synthesized? In 2.2 Authors wrote that urea was heated in crucible. The parameters indicate that it was heated on TG instrument (temperature increase of 10o/min) but the amount of sample indicate other method ? Normal oven? There is no information about this apparatus. Urea was heated in air or under argon (nitrogen)? Please explain.
  2. 2. Authors claim that adduct was immobilized on C3N4. But they simply evaporated the solvent. Was adduct really immobilized in 100%? It cannot be again dissolved at all or part dissolves? What part (10-90%). Please give additional information of this immobilization as this is the main subject of the manuscript. Was the product washed with something to remove not bonded part of complex? The measurements (as for example IR spectra) were done on purified sample or on simply evaporated solution with addition of C3N4? Without this data I must say that in manuscript there is no any proof that immobilization on C3N4 took place.
  3. 1a what solvent was used?
  4. 1. Why Bu-ChA have maximum at ca 505 nm (Fig. 1a) while in Fig. 1b at 570nm ? Why authors says that after mixing Bu with ChA absorption of complex is shifted to 550 nm and refer to Fig 1a, where this maximum is at ca 505nm?
  5. Why Bu-ChA in Fig. 1a do not have band at 370 nm (a minimum is observed in this region), while in Fig. 1b has? Are we observe two different compounds? Moreover band at 370 is much more intense that those at ca. 570 nm, thus in Fig. 1a should be also observed and is not. Why?
  6. The e of complex was wrongly calculated. If K = 4.2x103 than in Fig 1b absorption of the complex at 1.2 mM of ChA should be 35, while in Fig. 1b is < 0.1. Probably extinction coefficient is also too high (350.000 seems to be much to high). Authors study the adduct stabilized by weak hydrogen bonds, the high value of K seems to be much to high, I would expect K values much < 103. Please check all calculations and later please calculate the adduct concentration and its absorption in experimental conditions to check that everything fits.
  7. Authors used the Benesi-Hildebrand plots to calculate the K and e. In Fig. 1c it is well seen that this method do not work here properly. This method require that (in case presented in manuscript) [ChA] must be much higher than [Bu]. As during titration (Fig. 1b) [Bu] = 0.05 and [ChA] vary from 0 to 1.2 it is difficult to say that this condition is fulfilled, especially for first points. In such a case not 1 : 1 but also 2: 1 adduct can be observed (see literature) giving S shaped curves as seen in Fig. 1c. R = 0.96 also indicate that points do not give linear dependence (R should be > 0.98). All this gave problems with K and e determination, as pointed in my comments (point 6).
  8. Why in Fig. 2 b the final % of mass loss is negative? Please check experimental data. If this value was negative how the total mass change was calculated?
  9. 3. Please include two figures, one with 2500-4000 cm-1 and second with 500-2000 cm-1 range. Now it is very difficult to compare the spectra. I would suggest also to move this figures to supplementary file.
  10. Page 5, line 169. What means “However, the two HBs isomer (Fig. 5a) was more stable…” ? In Fig. 5a there is one adduct presented and not two of them, Maybe Authors would like to use “second” and not “two”?
  11. The hydrogen bonds in adduct are extremely weak (angles of 131o, 148o indicates this and this is the main criteria of bond strengths). Why Authors suggest that such a weak interactions promote adduct from fig. 5A and not that from Fig. 5b, where bond angle in 178o? This suggest that there is some problem with the calculations which do not agree with the bond length and angle. For such an extremely weak interaction why K is so high?? Why Authors used gas phase for their calculations and not the solvent?
  12. 6d, e and f. If the absorption of BuChA adduct changes slowly in time and reaches maximum after >55 min (in fig. 6 f, after 55 min plateau is still not observed) that after what time spectral changes were measured in Fig. 1? They should be measured after all spectral changes. Authors do not gave this detail. If spectra in Fig. 1 were measure after short time, the data are not correct as were measured during reaction (while Authors uses this data to calculate the equilibrium constant). Second problem is that Authors claim (page 7 line 204) that presence of C3N4 stabilize the anion formation and increases the reaction rate. At Fig. f rather decrease of rate is observed (especially for 5% C3N4 it is obvious); figs e and f indicate that C3N4 gave high background but absorbance change is higher in fig. d (from ca 0.01 to 0.12) than in the presence of C3N4 (change from 0.04 to 0.13). It indicates that presence of C3N4 destabilize the adduct formation.

Author Response

Open Review

English language and style

( ) Extensive editing of English language and style required
(x) Moderate English changes required
( ) English language and style are fine/minor spell check required
( ) I don't feel qualified to judge about the English language and style

Yes

Can be improved

Must be improved

Not applicable

Does the introduction provide sufficient background and include all relevant references?

(x)

( )

( )

( )

Is the research design appropriate?

( )

( )

(x)

( )

Are the methods adequately described?

( )

( )

(x)

( )

Are the results clearly presented?

( )

( )

(x)

( )

Are the conclusions supported by the results?

( )

( )

( )

(x)

Comments and Suggestions for Authors

Comments

In general paper contains information which could be published. In present form I have many questions as Authors didn’t include many crucial information. There is no any proof that adduct was immobilized and this a main aspect of the manuscript. Moreover presented data are sometimes in contrary to Authors claims and do not agree with presented spectra. Many aspects must be rethink by Authors and maybe some new calculations will be necessary. Thus in my opinion manuscript requires major revision before could be accepted. In present form I suggest to reject the manuscript. Authors should improve also the grammar and other language problems (for example $ 2.1, 3.1 and Fig. 1 captions starts with small letters).

Please give information which is missing:

  1. In what conditions C3N4 was synthesized? In 2.2 Authors wrote that urea was heated in crucible. The parameters indicate that it was heated on TG instrument (temperature increase of 10o/min) but the amount of sample indicate other method? Normal oven? There is no information about this apparatus. Urea was heated in air or under argon (nitrogen)? Please explain.

The g-C3N4 were prepared by heating urea in an oven and covered crucible. The required changes have been performed in the text and highlighted.

  1. 2. Authors claim that adduct was immobilized on C3N4. But they simply evaporated the solvent. Was adduct really immobilized in 100%? It cannot be again dissolved at all or part dissolves? What part (10-90%). Please give additional information of this immobilization as this is the main subject of the manuscript. Was the product washed with something to remove not bonded part of complex? The measurements (as for example IR spectra) were done on purified sample or on simply evaporated solution with addition of C3N4? Without this data I must say that in manuscript there is no any proof that immobilization on C3N4 took place.

The obtained solid product (x-g-C3N4/BUChA) was washed with ethanol for three times to remove non-bonded BUChA complex. The purified solid was further used for different characterization.

  1. 1a what solvent was used?

The immobilization of (x-g-C3N4/BUChA) were performed in methanol.

  1. 1. Why Bu-ChA have maximum at ca 505 nm (Fig. 1a) while in Fig. 1b at 570nm ? Why authors says that after mixing Bu with ChA absorption of complex is shifted to 550 nm and refer to Fig 1a, where this maximum is at ca 505nm?

Thank you for the reviewer comment. Fig. 1a was updated to show the BU-ChA spectrum, which shows the presence of the 570 nm peak.

  1. Why Bu-ChA in Fig. 1a do not have band at 370 nm (a minimum is observed in this region), while in Fig. 1b has? Are we observe two different compounds? Moreover band at 370 is much more intense that those at ca. 570 nm, thus in Fig. 1a should be also observed and is not. Why?

Fig. 1a was updated to show the full spectrum of Bu, ChA, and BU-ChA, which shows the presence of the 370 nm peak.

  1. The e of complex was wrongly calculated. If K = 4.2x103 than in Fig 1b absorption of the complex at 1.2 mM of ChA should be 35, while in Fig. 1b is < 0.1. Probably extinction coefficient is also too high (350.000 seems to be much to high). Authors study the adduct stabilized by weak hydrogen bonds, the high value of K seems to be much to high, I would expect K values much < 103. Please check all calculations and later please calculate the adduct concentration and its absorption in experimental conditions to check that everything fits.

The formation constant and extinction coefficient of the BuChA were recalculated using modified Benesi-Hildebrand equation. the calculated values were KCT and eCT were 2.8´103 and 2.4´103 respectively.

  1. Authors used the Benesi-Hildebrand plots to calculate the K and e. In Fig. 1c it is well seen that this method do not work here properly. This method require that (in case presented in manuscript) [ChA] must be much higher than [Bu]. As during titration (Fig. 1b) [Bu] = 0.05 and [ChA] vary from 0 to 1.2 it is difficult to say that this condition is fulfilled, especially for first points. In such a case not 1 : 1 but also 2: 1 adduct can be observed (see literature) giving S shaped curves as seen in Fig. 1c. R = 0.96 also indicate that points do not give linear dependence (R should be > 0.98). All this gave problems with K and e determination, as pointed in my comments (point 6).

The formation constant and extinction coefficient of the BuChA were recalculated using modified Benesi-Hildebrand equation.

  1. Why in Fig. 2 b the final % of mass loss is negative? Please check experimental data. If this value was negative how the total mass change was calculated?

           The increase of % sample mass loss is sometimes possible. This may be because

           of the reaction with gas environment, i.e. chemical reaction with gas or absorption

           of gas.

  1. 3. Please include two figures, one with 2500-4000 cm-1 and second with 500-2000 cm-1 range. Now it is very difficult to compare the spectra. I would suggest also to move this figures to supplementary file.

The required changes have been performed (see supporting information, Fig. S1 and S2).

  1. Page 5, line 169. What means “However, the two HBs isomer (Fig. 5a) was more stable…” ? In Fig. 5a there is one adduct presented and not two of them, Maybe Authors would like to use “second” and not “two”?

Thank you for the reviewer comment. The required changes have been performed.

  1. The hydrogen bonds in adduct are extremely weak (angles of 131o, 148o indicates this and this is the main criteria of bond strengths). Why Authors suggest that such a weak interactions promote adduct from fig. 5A and not that from Fig. 5b, where bond angle in 178o? This suggest that there is some problem with the calculations which do not agree with the bond length and angle. For such an extremely weak interaction why K is so high?? Why Authors used gas phase for their calculations and not the solvent?

The bond distance and bond angles were revised. The required changes have been included in the manuscript. The optimized parameters of BUChA-II gives short HB (bond distance 1.75 and bond angle 1530). The H10…Cl27 HB (bond distance 2.49 Å and bond angle 1480).  In addition, we calculated the stability of the complexes in methanol. The required changes have been included in the text and highlighted.

  1. 6d, e and f. If the absorption of BuChA adduct changes slowly in time and reaches maximum after >55 min (in fig. 6 f, after 55 min plateau is still not observed) that after what time spectral changes were measured in Fig. 1? They should be measured after all spectral changes. Authors do not gave this detail. If spectra in Fig. 1 were measure after short time, the data are not correct as were measured during reaction (while Authors uses this data to calculate the equilibrium constant).

We highly appreciate the comment. The formation constant of BuChA was measured after 60 min from mixing Bu and ChA solutions in order to minimize the time effect.

  1. Second problem is that Authors claim (page 7 line 204) that presence of C3N4 stabilize the anion formation and increases the reaction rate. At Fig. f rather decrease of rate is observed (especially for 5% C3N4 it is obvious); figs e and f indicate that C3N4 gave high background but absorbance change is higher in fig. d (from ca 0.01 to 0.12) than in the presence of C3N4 (change from 0.04 to 0.13). It indicates that presence of C3N4 destabilize the adduct formation.

We are totally agrees with the reviewer comment. The g-C3N4 sheets destabilize the adduct formation. The text is changes accordingly.

Reviewer 2 Report

The manuscript by Mersal et al. describes the charge transfer process between barbituric and chloroanilic moieties over g-C3N4. I think the topic of this manuscript may deserve publication in this journal. However, I have found some weaknesses and important gaps, and therefore, the current version is not suitable for publication and requires modifications. In this context, the authors should clarify and modify the following comments to improve the manuscript:

The main concern is regarding DFT calculations. Firstly, it should be desirable to explain why the isomer of barbituric acid was selected. There are 16 tautomers for barbituric acid and the most stable proved to be another one (i.e. Struct. Chem. 2014, 25, 1805). If the reason is its stability in solid state, please cite accordingly (Angew. Chem. Int. Ed. 2011, 50, 7924)

Hydrogen bond is a highly directional non-covalent interaction. The IUPAC definition stablishes that the angle X-H···Y is almost linear (ca. 180º). Therefore, it is quite odd that an angle of 148º (O7···H15-O14) provides the most stable BU···ChA complex. Is the value of 2.49 kcal/mol referred to a binding energy?

Following the same considerations, CH···O contact should be also carefully checked.

The values of electrostatic potential should be given in kcal/mol instead of eV. In addition, I have re-calculated the MEP surface of this tautomer and the value of V along the O11-H13 bond should be ca. 69 kcal/mol. Since this depends on the isodensity value, author should specify this point.

Minor points:

Some references suggest BA instead of BU for barbituric acid.

Page 4, line 141. Is there any reaction between barbituric acid and chloroanilic acid? In my opinion, authors should change “reaction” by interaction.

Figure 5b. This is another BA···ChA complex but not a BUChA “side view”

Author Response

Reviewer 2

Open Review

English language and style

( ) Extensive editing of English language and style required
( ) Moderate English changes required
(x) English language and style are fine/minor spell check required
( ) I don't feel qualified to judge about the English language and style

Yes

Can be improved

Must be improved

Not applicable

Does the introduction provide sufficient background and include all relevant references?

(x)

( )

( )

( )

Is the research design appropriate?

( )

(x)

( )

( )

Are the methods adequately described?

( )

( )

(x)

( )

Are the results clearly presented?

( )

( )

(x)

( )

Are the conclusions supported by the results?

( )

( )

(x)

( )

Comments and Suggestions for Authors

The manuscript by Mersal et al. describes the charge transfer process between barbituric and chloroanilic moieties over g-C3N4. I think the topic of this manuscript may deserve publication in this journal. However, I have found some weaknesses and important gaps, and therefore, the current version is not suitable for publication and requires modifications. In this context, the authors should clarify and modify the following comments to improve the manuscript:

The main concern is regarding DFT calculations. Firstly, it should be desirable to explain why the isomer of barbituric acid was selected. There are 16 tautomers for barbituric acid and the most stable proved to be another one (i.e. Struct. Chem. 2014, 25, 1805). If the reason is its stability in solid state, please cite accordingly (Angew. Chem. Int. Ed. 2011, 50, 7924).

We totally agree with the reviewer. We selected the most stable structure of BU for the DFT calculations. The text was changes and citing the required reference.

Hydrogen bond is a highly directional non-covalent interaction. The IUPAC definition stablishes that the angle X-H···Y is almost linear (ca. 180º). Therefore, it is quite odd that an angle of 148º (O7···H15-O14) provides the most stable BU···ChA complex. Is the value of 2.49 kcal/mol referred to a binding energy?

The bond distance and bond angles were revised. The required changes have been included in the manuscript. The optimized parameters of BUChA-II gives short HB (bond distance 1.75 and bond angle 1530). The H10…Cl27 HB (bond distance 2.49 Å  and bond angle 1480). 

Following the same considerations, CH···O contact should be also carefully checked.

The CH···O contacts reveals HB formations (C2H12…O18, 2.40 Å and 131° bond angle).

The values of electrostatic potential should be given in kcal/mol instead of eV. In addition, I have re-calculated the MEP surface of this tautomer and the value of V along the O11-H13 bond should be ca. 69 kcal/mol. Since this depends on the isodensity value, author should specify this point.

The values of MESP were given in kcal/mol. The required changes have been integrated into the text.

Minor points:

Some references suggest BA instead of BU for barbituric acid.

In this manuscript we uses BU for barbituric acid.

Page 4, line 141. Is there any reaction between barbituric acid and chloroanilic acid? In my opinion, authors should change “reaction” by interaction.

We agree with the reviewer. The required changes have been integrated into the manuscript.

Figure 5b. This is another BA···ChA complex but not a BUChA “side view”

The required changes have been performed.

Reviewer 3 Report

This is a combined experimental and computational study on the charge transfer process involving the charge transfer between barbituric acid and chloranilic acid in bulk and on a carbon nitrogen graphene nanosheet. On the premise that the English language is often unclear and must definitively be checked, I have focused mainly on the computational part which is my expertise and have important concerns. First, the computational detail section is extremely poor and the omission of computed structures with their energies and vibrational analysis in a supporting information file makes the data not reproducible which is not acceptable in a publication.

Then, the authors define suitable dimers on the basis of intermolecular hydrogen bonds but it seems that the mutual orientation is intentionally chosen and not the result of a systematic analysis which could be performed for example with tight binding approach and is definitely more appropriate. No details are provided about the nanosheet size and construction, neither in the text nor in the methods section. No solvatino effects are taken into account, limiting the discussion to the HOMO LUMO gap is indeed a limit. No TDDFT is carried out to better identify the charge transfer... the computational part is poor and seems pasted close to the experiment. 
Minor points:
the labels on the atoms are useless and difficult to read

different colors should be used for filled and empty MOs

Figure 7 is useless (and the edges are cut)

Author Response

Reviewer 3

Open Review

English language and style

(x) Extensive editing of English language and style required
( ) Moderate English changes required
( ) English language and style are fine/minor spell check required
( ) I don't feel qualified to judge about the English language and style

Yes

Can be improved

Must be improved

Not applicable

Does the introduction provide sufficient background and include all relevant references?

(x)

( )

( )

( )

Is the research design appropriate?

( )

(x)

( )

( )

Are the methods adequately described?

( )

( )

(x)

( )

Are the results clearly presented?

( )

( )

(x)

( )

Are the conclusions supported by the results?

( )

( )

(x)

( )

Comments and Suggestions for Authors

This is a combined experimental and computational study on the charge transfer process involving the charge transfer between barbituric acid and chloranilic acid in bulk and on a carbon nitrogen graphene nanosheet. On the premise that the English language is often unclear and must definitively be checked, I have focused mainly on the computational part which is my expertise and have important concerns.

First, the computational detail section is extremely poor and the omission of computed structures with their energies and vibrational analysis in a supporting information file makes the data not reproducible which is not acceptable in a publication.

Thank you for the comment. The required XY structures and the vibrational frequencies are included in supporting information section.

Then, the authors define suitable dimers on the basis of intermolecular hydrogen bonds but it seems that the mutual orientation is intentionally chosen and not the result of a systematic analysis which could be performed for example with tight binding approach and is definitely more appropriate.

We present here the most stable charge transfer complexes based on the binding energies.

No details are provided about the nanosheet size and construction, neither in the text nor in the methods section.

Thank you for the reviewer comment. The required data were included in the computational section. The g-C3N4 nanosheet were constructed based on the (011) lattice plane super cell (2 x 2 x 1) according to literature [48]. The terminal N atoms were truncated with H atoms for layer stability [49].

No solvation effects are taken into account, limiting the discussion to the HOMO LUMO gap is indeed a limit. No TDDFT is carried out to better identify the charge transfer... the computational part is poor and seems pasted close to the experiment.

The solvent effects on the formed charge transfer complexes were performed using self-consistent reaction field (SCRF) method and polarized continuum model (PCM). The method details were added to the computational section, while the results were integrated into the text. The charge transfer process was investigated using Hirshfeld atomic population charges models. The quantative charge transfer analysis confirms the higher stability of BUChA-I than BUChA-II.

Minor points:
the labels on the atoms are useless and difficult to read

The figures were updated with high resolution.

Different colors should be used for filled and empty MOs.

We tries different color for the MOs, however, the current color for MOs gives the better contrast.

Figure 7 is useless (and the edges are cut).

We are presenting Figure 7a to show the p-p stacking interactions and vdW forces, which stabilizes the composite. In Fig. 7b, we highlighted the Figure to show the bond distance and atoms labeling.

Round 2

Reviewer 1 Report

In general Authors responsed to all my questions and suggestions. I din't expect that they will be able to do that so fast and in proper way.  The problem with immobilization was explained, the K and e values were recalculated and now they are in much  more resonable range. Still I think that some measurements could be done in different way, but I think that after this great improvement manuscript can be accepted in present form.

Author Response

Thank you for your time and help

Reviewer 2 Report

In my opinion authors have addressed my previous concerns. The manuscript deserves its publication in the present form

Author Response

Thank you for your time and help

Reviewer 3 Report

The authors have improved the first version of the manuscript, although they have not exactly replied to my concerns. First, the dimer structures. When they state that they are built based on the binding energies, I again reply that a systematic search should have been carried out. They have not explained how the most stable structures were found. How many structures have they tried? Which rule did they follow to build trial structures? This is why I had recommended a systematic search.

Then the second major concern was the assessment of the charge transfer based on the charges. TDDFT calculations would be more appropriate, but also in this case the authors have not considered my comment. This shiuld be at least justified in the text, because TDDFT is expected and is not found.

Author Response

The authors have improved the first version of the manuscript, although they have not exactly replied to my concerns. First, the dimer structures. When they state that they are built based on the binding energies, I again reply that a systematic search should have been carried out. They have not explained how the most stable structures were found. How many structures have they tried? Which rule did they follow to build trial structures? This is why I had recommended a systematic search.

Thank you for the reviewer comment and revision. The charge transfer complexe between BU and ChA could be occurs through hydrogen bonding between the terminal OH and C=O of BU with similar groups on ChA (I), the hydrogen bonding with the terminal OH and C=O of BU with Cl and OH terminal of ChA (II), the T-shaped structure between BU and ChA (III), and p-p forces, where BU were in top of the ChA (IV). We explored the optimized structures of the four possible orientations. The results are presented in Fig. 5 and Fig. S3 and discussed in the text.

Then the second major concern was the assessment of the charge transfer based on the charges. TDDFT calculations would be more appropriate, but also in this case the authors have not considered my comment. This shiuld be at least justified in the text, because TDDFT is expected and is not found.

The required calculations have been performed and included in the text (Fig. S4). The results were discussed in page 9-10).
